# UNSUPERVISED ORDER LEARNING

**Seon-Ho Lee, Nyeong-Ho Shin & Chang-Su Kim**
School of Electrical Engineering, Korea University
Seoul 02841, Korea
{seonholee,nhshin}@mcl.korea.ac.kr, changsukim@korea.ac.kr

## ABSTRACT

A novel clustering algorithm for orderable data, called unsupervised order learning (UOL), is proposed in this paper. First, we develop the ordered $k$-means to group objects into ordered clusters by reducing the deviation of an object from consecutive clusters. Then, we train a network to construct an embedding space, in which objects are sorted compactly along a chain of line segments, determined by the cluster centroids. We alternate the clustering and the network training until convergence. Moreover, we perform unsupervised rank estimation via a simple nearest neighbor search in the embedding space. Extensive experiments on various orderable datasets demonstrate that UOL provides reliable ordered clustering results and decent rank estimation performances with no supervision. The source codes are available at https://github.com/seon92/UOL.

## 1 INTRODUCTION

Clustering aims to partition objects into clusters by finding underlying object classes without any supervision. It is a fundamental task with multifarious applications, including anomaly detection (Emadi & Mazinani, 2018), domain adaptation (Tang et al., 2020), community detection (Xing et al., 2022), and representation learning (Yang et al., 2016). Especially, we can now collect countless images easily, but it is both time-consuming and expensive to manually annotate such images. Unsupervised clustering can reduce the annotation burden by providing initial predictions.

Many deep clustering methods (Caron et al., 2018; Van Gansbeke et al., 2020; Tao et al., 2021; Tsai et al., 2021; Huang et al., 2022) have been developed to provide promising results based on deep learning. However, most of them are for clustering nominal data for classification (Deng et al., 2009) or segmentation (Ji et al., 2019), so they may yield suboptimal results on orderable data.

Orderable data — used in many applications such as medical assessment (Yang et al., 2021), facial age estimation (Ricanek & Tesafaye, 2006), and aesthetic quality assessment (Kong et al., 2016) — are different from nominal data in that objects can be arranged in an intrinsic order (Lim et al., 2020). For example, medical images can be sorted according to the severity of illness, and facial photos according to the subjects' ages. Compared to nominal data, clustering of orderable data is more challenging because it is difficult to distinguish adjacent underlying classes; in facial photos, a 20-year-old tends to be very similar to a 19-year-old or a 21-year-old.

Despite its difficulty, the clustering of orderable data has many benefits, just as the clustering of nominal data does; it can facilitate an initial understanding of data characteristics. Also, in general, it is costly to obtain reliable annotations for orderable data due to the aforementioned ambiguity between classes. The clustering of orderable data can reduce the annotation cost, *e.g.* for medical assessment datasets.

In this paper, we propose a novel algorithm, called unsupervised order learning (UOL), for clustering orderable data. It aims to discover underlying classes of objects as well as their ordering relationship without any supervision.

- To this end, we first develop the ordered $k$-means by extending the ordinary $k$-means (Gersho & Gray, 1991). Whereas the ordinary $k$-means minimizes the distance of an object to its centroid, the ordered $k$-means reduces the deviation of an object from consecutive centroids, thereby laying out adjacent clusters close to each other in an embedding space. Hence, as illustrated in Figure 1(a), the clusters of the ordered $k$-means form a clear sequential order.

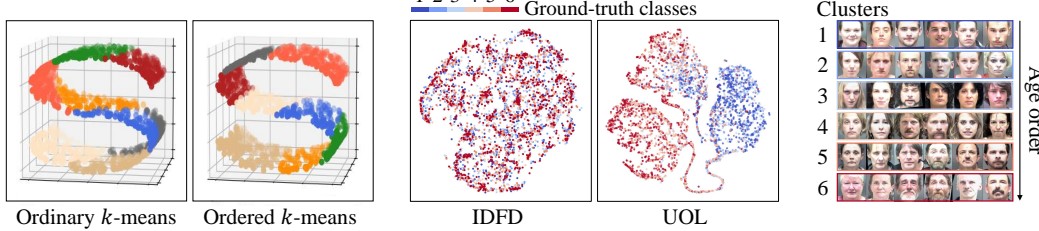

(a) Comparison of clustering results  (b) Comparison of embedding spaces  (c) UOL clustering examples

Figure 1: (a) Comparison of the ordinary $k$-means and the ordered $k$-means on data samples around an S curve in the 3D space. (b) Comparison of the embedding spaces for dividing the MORPH II dataset (Ricanek & Tesafaye, 2006) into six clusters, obtained by IDFD (Tao et al., 2021) and the proposed UOL algorithm. We group the ground-truth age classes into six age ranges, depicted by colors from blue to red. (c) Examples in each cluster of UOL on the MORPH II dataset.

- Furthermore, to design an even better space for ordered clustering, we train an embedding network based on the order desideratum that encourages objects to be arranged according to their cluster indices. We iterate the ordered $k$-means clustering and the network training alternately. For example, in Figure 1(b), the proposed UOL well arranges object instances in a facial dataset, MORPH II (Ricanek & Tesafaye, 2006), according to their ordered classes in the embedding space, while the deep clustering algorithm IDFD (Tao et al., 2021) locates the instances in different classes closely without distinction. Consequently, UOL discovers the underlying age order with no supervision, as shown in Figure 1(c). In other words, given only facial images, UOL sorts them roughly according to the ages.

Extensive experiments demonstrate that UOL provides reliable clustering results on various orderable datasets and also yields promising rank estimation results with no supervision.

The contributions of this paper can be summarized as follows.

- We propose the *first* deep clustering algorithm for orderable data, which discovers clusters together with their hidden order relationships.
- We develop the ordered $k$-means by extending the ordinary $k$-means. It not only groups object instances effectively but also sorts the clusters to form a meaningful order. We also prove the local optimality of the solution.
- The proposed algorithm provides promising clustering results on various types of orderable data for facial age estimation, medical assessment, facial expression recognition, and speech evaluation. Also, it shows decent estimation results as an unsupervised rank estimator.

## 2 RELATED WORK

**Deep clustering:** In deep clustering, instances are grouped into clusters while a network is trained to embed them into a better space for clustering. Most deep clustering algorithms (Xie et al., 2016; Chang et al., 2017; Van Gansbeke et al., 2020; Wu et al., 2019; Yang et al., 2016; Tsai et al., 2021; Tao et al., 2021; Huang et al., 2022; Niu et al., 2022) adopt the alternate learning framework; they repeat clustering and network training alternately at regular intervals. Caron et al. (2018) employed $k$-means to cluster instances and trained a network to classify instances simply according to their cluster memberships. Tao et al. (2021) also used $k$-means, but they trained a network with a feature decorrelation constraint to construct a better embedding space. Recently, Van Gansbeke et al. (2020); Huang et al. (2022); Niu et al. (2020); Ji et al. (2019); Tsai et al. (2021) have adopted the contrastive loss for network training, as self-supervised learning techniques (Chen et al., 2020; He et al., 2020) have shown promising results for constructing discriminative embedding spaces.

Even though many deep clustering algorithms have been proposed, most of them are designed for ordinary data. Thus, when applied to orderable data, they may yield sub-optimal clusters because there is no clear distinction between classes in orderable data. In contrast, the proposed algorithm provides more reliable clustering results by considering the order — which is unknown but to be discovered — of underlying classes.

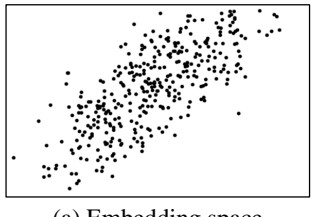
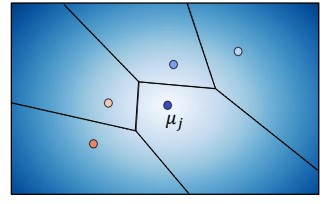
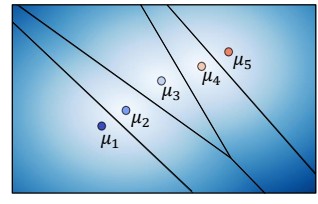

(a) Embedding space      (b) Ordinary $k$-means      (c) Ordered $k$-means

Figure 2: Instances in an embedding space are clustered by the ordinary $k$-means and the ordered $k$-means, respectively, at $k = 5$. In (b) and (c), each circle represents the centroid of a cluster. In (b), each point $h_x$ in the embedding space is colorized according to its distance $d(h_x, \mu_j)$ from centroid $\mu_j$. In (c), $h_x$ is colorized according to the deviation $\delta(h_x, \mu_{j_p}, \mu_j, \mu_{j_n})$ in equation 1. Here, $j = 3$.

Also, there are ordinal clustering algorithms (Zhang & Cheung, 2020; Jia & Cheung, 2017; Zhang & Cheung, 2018), but ordinal clustering is different from the proposed ordered clustering. Ordinal clustering aims to group instances with provided ordinal attributes, whereas ordered clustering, without any supervision, aims to discover clusters together with their meaningful order.

**Orderable data:** There are various types of orderable data. For example, in a video-sharing platform, videos can be sorted according to the number of views or likes. Similarly, medical images, aesthetic images, and facial images can be arranged according to illness severity, scores, or ages, respectively. If the intrinsic order in orderable data is known, they are called ordered data, in which classes (or ranks) form an ordered set. To estimate the classes of ordered data, several attempts (Lim et al., 2020; Lee & Kim, 2021; Shin et al., 2022; Lee et al., 2022; Lee & Kim, 2022) based on order learning have been proposed. Lim et al. (2020) first proposed the notion of order learning, which learns ordering relationships between objects using a pairwise comparator and determines the rank of an unseen object by comparing it with references. Lee & Kim (2021) improved the performance of order learning by finding reliable references via order-identity feature decomposition. Shin et al. (2022) proposed a regression approach for order learning. Lee et al. (2022) simplified order learning; their method sorts objects in an embedding space and then uses the $k$-NN search to estimate the rank of an object, instead of comparing it with references. Also, Lee & Kim (2022) developed a weakly supervised training scheme for order learning.

Note that, in (Lim et al., 2020; Lee & Kim, 2021), ordered data are clustered according to order-unrelated properties, whereas we discover an underlying order in orderable data and cluster the data according to the discovered order. It is worth pointing out that all existing order learning algorithms assume that the ground-truth ranks of training objects are completely or partially known. In contrast, we attempt to cluster orderable data $\mathcal{X}$ and estimate the rank of each instance in $\mathcal{X}$ without any supervision. In other words, we propose the *first* unsupervised algorithm for order learning.

## 3 PROPOSED ALGORITHM

### 3.1 ORDERED $k$-MEANS

Suppose that there are $m$ instances in a training set $\mathcal{X} = \{x_1, x_2, \ldots, x_m\}$. In orderable data, classes can be arranged in a natural order. Hence, the proposed ordered $k$-means aims to partition $\mathcal{X}$ into $k$ clusters $\{\mathcal{C}_j\}_{j=1}^k$ where $\mathcal{C}_1 \to \mathcal{C}_2 \to \cdots \to \mathcal{C}_k$ represents a meaningful order.

An encoder $h$ maps each instance $x \in \mathcal{X}$ into a feature vector $h_x = h(x)$ in an embedding space. As $h$, we adopt VGG16 (Simonyan & Zisserman, 2015) without fully connected layers. To measure the dissimilarity in the embedding space, we use the Euclidean distance $d$. Let $\mu_j$ denote the 'centroid' or the representative vector for the instances in cluster $\mathcal{C}_j$. Then, these centroids form a chain $\mu_1 \to \mu_2 \to \cdots \to \mu_k$ in the embedding space, as in Figure 2(c). Note that a chain is defined as a maximal linearly-ordered set (Lim et al., 2020). For $x \in \mathcal{C}_j$, the deviation of $x$ from the chain is defined as

$$\delta(h_x, \mu_{j_p}, \mu_j, \mu_{j_n}) = d^2(h_x, \mu_j) + \alpha d^2(h_x, \mu_{j_p}) + \alpha d^2(h_x, \mu_{j_n}) \tag{1}$$

where $\alpha$ is a positive weight, and $j_p$ and $j_n$ are the indices for the previous and next clusters;

$$j_p = \begin{cases} j-1 & \text{if } j \neq 1 \\ j+1 & \text{if } j = 1 \end{cases}, \qquad j_n = \begin{cases} j+1 & \text{if } j \neq k \\ j-1 & \text{if } j = k \end{cases}. \tag{2}$$

---

**Algorithm 1** Unsupervised Order Learning (UOL)

---

**Input:** Data $\mathcal{X} = \{x_1, x_2, \ldots, x_m\}$, $k$ = the number of clusters

 1: **repeat**
 2:   Randomly partition $\mathcal{X}$ into $\mathcal{C}_1, \mathcal{C}_2, \ldots, \mathcal{C}_k$
 3:   **repeat**
 4:    **for all** $j = 1, 2, \ldots, k$ **do**
 5:     Update centroid $\mu_j$ via equation 4            ▷ Centroid rule
 6:    **end for**
 7:    **for all** $j = 1, 2, \ldots, k$ **do**
 8:     Update cluster $\mathcal{C}_j$ via equation 5           ▷ NCC rule
 9:    **end for**
10:    Permute the cluster indices via equation 6       ▷ Permutation rule
11:   **until** convergence or predefined number of iterations
12:   Fine-tune the encoder $h$ and the centroids $\{\mu_j\}_{j=1}^k$ to minimize $\ell_{\text{total}}$ in equation 11
13: **until** predefined number of epochs

**Output:** Clusters $\{\mathcal{C}_j\}_{j=1}^k$, centroids $\{\mu_j\}_{j=1}^k$, encoder $h$

---

In the case of the first cluster $\mathcal{C}_1$, there is no previous cluster. Thus, by defining $j_{\mathrm{p}} = j + 1$, the deviation in equation 1 becomes $d^2(h_x, \mu_j) + 2\alpha d^2(h_x, \mu_{j_{\mathrm{n}}})$ so that it is balanced with those for middle clusters. Similarly, for the last cluster $\mathcal{C}_k$, $j_{\mathrm{n}} = j - 1$.

In equation 1, the first term measures the distance of an instance in $\mathcal{C}_j$ to its centroid $\mu_j$ as in the ordinary $k$-means (Gersho & Gray, 1991). In the ordered $k$-means, however, the second and third terms further consider the distances to the adjacent centroids. This is because, in orderable data, adjacent clusters contain similar instances in general. For example, in facial photos, a 20-year-old tends to be similar to a 19-year-old or a 21-year-old. Therefore, instances in $\mathcal{C}_j$ should be closer to the adjacent clusters $\mathcal{C}_{j_{\mathrm{p}}}$ and $\mathcal{C}_{j_{\mathrm{n}}}$ than to the other clusters. Moreover, in orderable data, instances vary continuously between adjacent clusters in general. For example, although discrete age classes are defined, there are 19.5-year-olds. Thus, the deviation in equation 1 implies that an instance in $\mathcal{C}_j$ should be concentrated near the line segments $\mu_{j_{\mathrm{p}}} \rightarrow \mu_j \rightarrow \mu_{j_{\mathrm{n}}}$, as illustrated by the color map in Figure 2(c), not just around the centroid $\mu_j$.

Next, we measure the overall quality of ordered clustering by

$$J(\{\mathcal{C}_i\}_{i=1}^k, \{\mu_i\}_{i=1}^k) = \sum_{j=1}^k \sum_{x \in \mathcal{C}_j} \delta(h_x, \mu_{j_{\mathrm{p}}}, \mu_j, \mu_{j_{\mathrm{n}}}). \tag{3}$$

We attempt to find the optimum clusters and centroids to minimize this objective function $J$ in equation 3, yet finding the global optimum is NP-complete (Garey et al., 1982). Hence, we propose an iterative algorithm to find a local optimum, as in the ordinary $k$-means (Gersho & Gray, 1991).

**Centroid rule:** After fixing the clusters $\{\mathcal{C}_j\}_{j=1}^k$, we update the centroids $\{\mu_j\}_{j=1}^k$ to minimize $J$ in equation 3. For $j \notin \{1, 2, k-1, k\}$, the optimal centroids are obtained as

$$\mu_j = \frac{\sum_{x \in \mathcal{C}_j} h_x + \alpha \sum_{x \in \mathcal{C}_{j_{\mathrm{p}}}} h_x + \alpha \sum_{x \in \mathcal{C}_{j_{\mathrm{n}}}} h_x}{|\mathcal{C}_j| + \alpha |\mathcal{C}_{j_{\mathrm{p}}}| + \alpha |\mathcal{C}_{j_{\mathrm{n}}}|}. \tag{4}$$

Note that centroid $\mu_j$ is given by a weighted average of the instances in the three consecutive clusters $\mathcal{C}_{j_{\mathrm{p}}}, C_j$, and $\mathcal{C}_{j_{\mathrm{n}}}$. For $j \in \{1, 2, k-1, k\}$, the centroid rule in equation 4 is slightly different due to endpoint effects. The full centroid rule is derived in Appendix B.1.

**NCC rule:** On the other hand, after fixing the centroids, we update the membership of each instance to minimize $J$ in equation 3. As detailed in Appendix B.2, the optimal cluster $\mathcal{C}_j$ is given by

$$C_j = \big\{x \,|\, \delta(h_x, \mu_{j_{\mathrm{p}}}, \mu_j, \mu_{j_{\mathrm{n}}}) \leq \delta(h_x, \mu_{l_{\mathrm{p}}}, \mu_l, \mu_{l_{\mathrm{n}}}) \text{ for all } 1 \leq l \leq k\big\}. \tag{5}$$

Note that the deviation $\delta(h_x, \mu_{j_{\mathrm{p}}}, \mu_j, \mu_{j_{\mathrm{n}}})$ in equation 1 computes the distances of $h_x$ to three consecutive centroids. We assign an instance to $\mathcal{C}_j$ if its nearest consecutive centroids (NCCs) are $(\mu_{j_{\mathrm{p}}}, \mu_j, \mu_{j_{\mathrm{n}}})$.

**Permutation rule:** The cluster indices should represent a meaningful order. Hence, after fixing the clusters and the centroids, we apply the permutation rule to rearrange the clusters optimally. Let

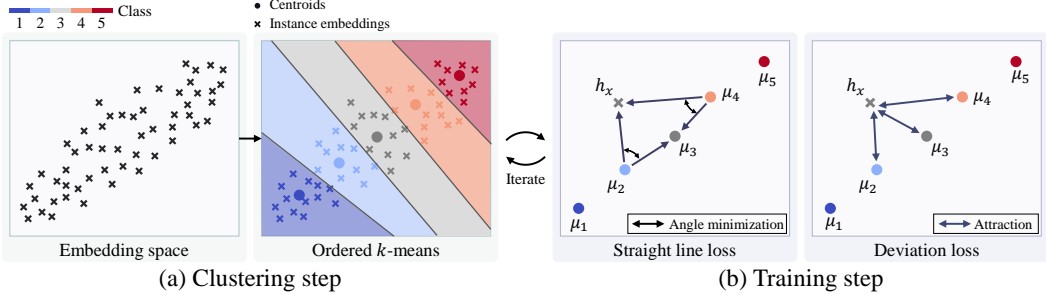

(a) Clustering step            (b) Training step

Figure 3: An overview of the proposed UOL algorithm.

$\sigma : \{1, \dots, k\} \to \{1, \dots, k\}$ be a bijective permutation function. Then, the optimal permutation $\sigma^*$ is determined to minimize $J$ in equation 3,

$$\sigma^* = \arg\min_{\sigma} J(\{\mathcal{C}_{\sigma(i)}\}_{i=1}^k, \{\mu_{\sigma(i)}\}_{i=1}^k). \tag{6}$$

To find the optimal permutation $\sigma^*$, $\frac{k!}{2}$ cases should be considered, which may be too demanding. Therefore, for a large $k$, we develop a greedy scheme for finding $\sigma$ with $O(k^3)$ complexity, which is described in Appendix B.3.

We apply the centroid rule, the NCC rule, and the permutation rule iteratively until convergence. Since all three rules monotonically decrease the same objective function $J$ in equation 3, it is guaranteed that the ordered $k$-means converges to a local minimum.

If $\alpha = 0$ in equation 1, the objective function $J$ in equation 3 is equal to that of the ordinary $k$-means (Gersho & Gray, 1991). In such a case, the resultant clusters are not ordered in general as illustrated in Figure 2(b), for the relations between adjacent clusters are not taken into account. In contrast, the proposed ordered $k$-means minimizes the average dissimilarity of the instances to their NCCs, thereby making consecutive clusters close to each other as shown in Figure 2(c).

### 3.2 UNSUPERVISED ORDER LEARNING (UOL)

We develop the UOL algorithm based on the ordered $k$-means. Figure 3 is an overview of UOL. Note that, in a given embedding space, the ordered $k$-means is applied to data $\mathcal{X}$ to yield clusters $\{\mathcal{C}_j\}_{j=1}^k$. Then, in turn, we fine-tune the encoder or equivalently revamp the embedding space, so the clusters are better arranged according to their order relationship. In ordered clustering, cluster indices represent their order, and the index difference between two clusters should be proportional to the dissimilarity between them. Hence, we fine-tune the encoder $h$ with the following desideratum:

$$i < j < l \quad \Leftrightarrow \quad d(h_x, h_y) < d(h_x, h_z) \tag{7}$$

where instances $x \in \mathcal{C}_i$, $y \in \mathcal{C}_j$, and $z \in \mathcal{C}_l$. In the fine-tuning process, we also optimize the centroids $\{\mu_j\}_{j=1}^k$ as learnable parameters to guide the cluster regions in the embedding space.

**Straight line loss:** To encourage the order desideratum in equation 7, we employ the straight line (SL) loss. First, let us define the direction vector $v(r, s)$ from point $r$ to point $s$ in the embedding space as

$$v(r, s) = (s - r)/\|s - r\|. \tag{8}$$

Then, we define the SL loss as

$$\ell_{\text{SL}} = \sum_{j=1}^k \sum_{x \in \mathcal{C}_j} \left(1 - v(\mu_{j_{\text{p}}}, h_x)^t v(\mu_{j_{\text{p}}}, \mu_j)\right) + \left(1 - v(\mu_{j_{\text{n}}}, h_x)^t v(\mu_{j_{\text{n}}}, \mu_j)\right). \tag{9}$$

To minimize the first term, the angle between $v(\mu_{j_{\text{p}}}, h_x)$ and $v(\mu_{j_{\text{p}}}, \mu_j)$ should be zero, as illustrated in Figure 3(b). In other words, $\mu_{j_{\text{p}}}$, $h_x$, and $\mu_j$ should be on a line in the embedding space. Similarly, due to the second term, $\mu_{j_{\text{n}}}$, $h_x$, and $\mu_j$ should be on the same line. Therefore, $\ell_{\text{SL}}$ encourages that $h_x$ and its NCCs $(\mu_{j_{\text{p}}}, \mu_j, \mu_{j_{\text{n}}})$ are arranged linearly on a 1D manifold in the embedding space, and that $h_x$ and $\mu_j$ are located between $\mu_{j_{\text{p}}}$ and $\mu_{j_{\text{n}}}$. In other words, it helps instances and centroids to be

Table 1: Comparison of ordered clustering results in settings A and B of MORPH II. The best results are boldfaced, and the second-best ones are underlined.

| | Setting A | | | | | | Setting B | | | | | |
|---|---|---|---|---|---|---|---|---|---|---|---|---|
| | SRCC($\uparrow$) | | | MAE ($\downarrow$) | | | SRCC ($\uparrow$) | | | MAE ($\downarrow$) | | |
| Algorithm | $k=6$ | $k=9$ | $k=12$ | $k=6$ | $k=9$ | $k=12$ | $k=6$ | $k=9$ | $k=12$ | $k=6$ | $k=9$ | $k=12$ |
| DAC (Chang et al., 2017) | 0.180 | 0.216 | 0.108 | 1.785 | 2.734 | 3.853 | 0.117 | 0.147 | 0.350 | 1.792 | 2.748 | 3.417 |
| DeepCluster (Caron et al., 2018) | 0.139 | 0.107 | 0.098 | 2.078 | 2.871 | 4.000 | 0.051 | 0.049 | 0.062 | 2.185 | 3.075 | 4.343 |
| IIC (Ji et al., 2019) | 0.044 | 0.135 | 0.332 | 2.312 | 2.959 | 3.420 | 0.112 | 0.120 | 0.109 | 1.828 | 2.672 | 3.868 |
| MiCE (Tsai et al., 2021) | 0.260 | 0.254 | 0.409 | 1.568 | 2.579 | 2.935 | 0.271 | 0.292 | 0.339 | 1.643 | 2.683 | 3.372 |
| IDFD (Tao et al., 2021) | 0.123 | 0.136 | 0.274 | 1.961 | 2.867 | 3.171 | 0.094 | 0.178 | 0.160 | 1.881 | 2.586 | 3.787 |
| ProPos (Huang et al., 2022) | 0.167 | 0.144 | 0.196 | 1.755 | 2.793 | 3.590 | 0.129 | 0.139 | 0.202 | 1.795 | 2.775 | 3.686 |
| Proposed UOL | **0.632** | **0.506** | **0.495** | **1.122** | **2.093** | **2.572** | **0.531** | **0.473** | **0.437** | **1.278** | **2.114** | **3.025** |

Figure 4: Examples of clustering results in setting A of MORPH II: (a) DeepCluster (Caron et al., 2018), (b) MiCE (Tsai et al., 2021), (c) IDFD (Tao et al., 2021), (d) ProPos (Huang et al., 2022), and (e) UOL. For each cluster, the average ground-truth age of its instances is also reported.

sorted according to the index order of the clusters along the 1D manifold in the high-dimensional embedding space, thereby satisfying the order desideratum in equation 7.

**Deviation loss:** In addition to $\ell_{\text{SL}}$, we employ the deviation loss $\ell_{\text{D}}$, which aims at reducing the deviation $\delta$ of an instance from its NCCs in equation 1, given by

$$\ell_{\text{D}} = \sum_{j=1}^{k} \sum_{x \in \mathcal{C}_j} \max(\delta(h_x, \mu_{j_{\text{p}}}, \mu_j, \mu_{j_{\text{n}}}) - \gamma, 0) \tag{10}$$

where $\gamma$ is a threshold.

Finally, we use the total loss function

$$\ell_{\text{total}} = \ell_{\text{SL}} + \ell_{\text{D}} \tag{11}$$

to optimize the encoder parameters and the centroids. **Algorithm 1** summarizes the proposed UOL.

### 3.3 UNSUPERVISED RANK ESTIMATION

We can estimate the rank (or class) of an unseen test instance based on the $K$-NN classification. We first transform a test instance $x$ to the embedded vector $h(x)$. Then, in the embedding space, we find the set $\mathcal{N}$ of its $K$ NNs among all training instances in $\mathcal{X}$. We then estimate the rank of $x$ by

$$\hat{\theta}(x) = \frac{1}{K} \sum_{y \in \mathcal{N}} \sum_{j=1}^{k} j[y \in \mathcal{C}_j] \tag{12}$$

where $[\cdot]$ is the indicator function.

## 4 EXPERIMENTS

This section provides various experimental results. More results are available in Appendix C.

Table 2: Comparison of ordered clustering results on the CLAP2015 dataset.

Table 3: Comparison of ordered clustering results on the DR and RetinaMNIST datasets at $k = 5$.

| | SRCC($\uparrow$) | | MAE ($\downarrow$) | |
|---|---|---|---|---|
| Algorithm | $k = 6$ | $k = 9$ | $k = 6$ | $k = 9$ |
| DAC (Chang et al., 2017) | 0.100 | 0.196 | 2.031 | 2.615 |
| DeepCluster (Caron et al., 2018) | 0.018 | 0.042 | 2.186 | 3.231 |
| IIC (Ji et al., 2019) | 0.123 | 0.102 | 1.892 | 2.900 |
| MiCE (Tsai et al., 2021) | 0.194 | 0.198 | 1.794 | 2.709 |
| IDFD (Tao et al., 2021) | 0.209 | 0.172 | 1.733 | 2.715 |
| ProPos (Huang et al., 2022) | 0.030 | 0.088 | 1.928 | 2.875 |
| Proposed UOL | **0.302** | **0.295** | **1.645** | **2.496** |

| | DR | | RetinaMNIST | |
|---|---|---|---|---|
| Algorithm | SRCC ($\uparrow$) | MAE ($\downarrow$) | SRCC ($\uparrow$) | MAE ($\downarrow$) |
| DAC (Chang et al., 2017) | 0.027 | 1.502 | 0.348 | 1.087 |
| DeepCluster (Caron et al., 2018) | 0.099 | 1.295 | 0.037 | 1.394 |
| IIC (Ji et al., 2019) | 0.205 | 1.284 | 0.165 | 1.346 |
| MiCE (Tsai et al., 2021) | 0.277 | 1.305 | 0.495 | 1.002 |
| IDFD (Tao et al., 2021) | 0.018 | 1.531 | 0.066 | 1.401 |
| ProPos (Huang et al., 2022) | 0.085 | 1.490 | 0.041 | 1.469 |
| Proposed UOL | **0.333** | **1.219** | **0.567** | **0.953** |

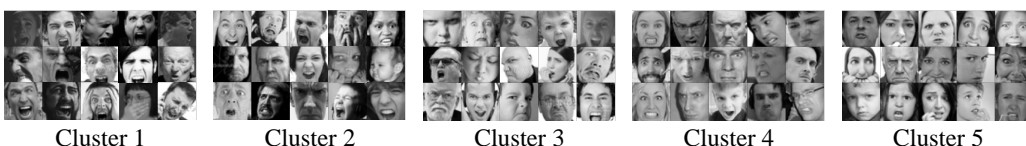

| Cluster 1 | Cluster 2 | Cluster 3 | Cluster 4 | Cluster 5 |

Figure 5: Examples of clustering results on the FER+ dataset at $k = 5$.

## 4.1 IMPLEMENTATION

We initialize the encoder $h$ with VGG16 pre-trained on ILSVRC2012 (Deng et al., 2009). We use the Adam optimizer (Kingma & Ba, 2015) with a batch size of 32 and a weight decay of $5 \times 10^{-4}$. We set the learning rate to $10^{-4}$. For data augmentation, we do random horizontal flips and random crops. More implementation details are in Appendix C.1.

## 4.2 DATASETS

**MORPH II** (Ricanek & Tesafaye, 2006): It is a dataset for age estimation, composed of 55,134 facial images in the age range $[16, 77]$. It provides age, gender, and race labels.

**CLAP2015** (Escalera et al., 2015): It contains 4,691 facial photos taken in various environments. For each photo, it provides an age label in the age range $[3, 85]$.

**DR** (Dugas et al., 2015): It is a dataset for diabetic retinopathy (DR) diagnosis. It provides 35,125 eye images. Each image is annotated with a 5-scale severity score.

**RetinaMNIST** (Yang et al., 2021): It is a small dataset for DR diagnosis. It contains 1,600 eye images, annotated with severity scores in a 5-scale.

**FER+** (Barsoum et al., 2016): It contains 32,298 grayscale images for facial expression recognition. Each image is categorized into one of eight emotion classes.

## 4.3 CLUSTERING

**Metrics:** In orderable data, classes can form an order. Without loss of generality, let us assume that the ordered classes are the first $k$ natural numbers. For instances $x, y,$ and $z$, suppose that $\theta(x) = i, \theta(y) = j,$ and $\theta(z) = l$, where $\theta(\cdot)$ is a class function and $i < j < l$. Then, misclustering $x$ with $z$ is a severer error than misclustering it with $y$. Thus, to evaluate the clustering performance of orderable data, the order between classes should be considered. We hence use the Spearman's rank correlation coefficient (SRCC) and mean absolute error (MAE) metrics. Specifically, SRCC is the correlation coefficient between two rankings, given by

$$\text{SRCC} = 1 - \frac{6 \sum_{j=1}^{k} \sum_{x \in \mathcal{C}_j} (j - \theta(x))}{k(k^2 - 1)}. \tag{13}$$

MAE is the average absolute error, defined as

$$\text{MAE} = \frac{1}{|\mathcal{X}|} \sum_{j=1}^{k} \sum_{x \in \mathcal{C}_j} |j - \theta(x)|. \tag{14}$$

**Comparison on age estimation datasets:** Table 1 compares the ordered clustering results in settings A and B of MORPH II. As in (Tsai et al., 2021; Tao et al., 2021; Caron et al., 2018), we assume that the number $k$ of clusters is known *a priori*. Also, according to $k$, the entire age classes are mapped into $k$ age groups. All algorithms in Table 1 use VGG16 pre-trained on ILSVRC2012 (Deng et al., 2009) as their encoder backbones. We train these methods using the official source codes.

As done in (Tsai et al., 2021; Tao et al., 2021; Caron et al., 2018), for the conventional algorithms, the obtained clusters are permuted via the Hungarian method (Kuhn, 1955) to match the ground-truth

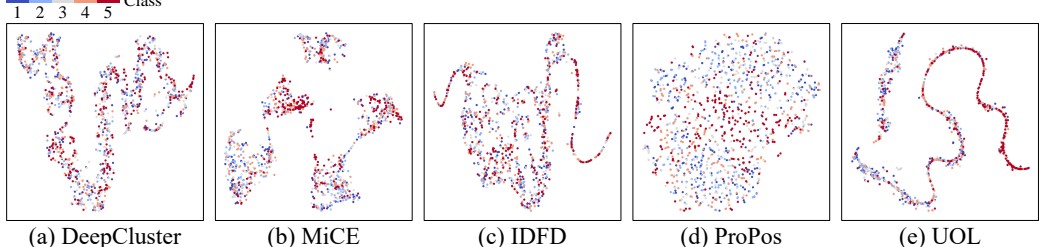

Figure 6: t-SNE visualization of the embedding spaces for the DR dataset at $k = 5$.

Table 4: Comparison of rank estimation results (MAEs) on the MORPH II, DR, and RetinaMNIST datasets. For MORPH II, $k = 6$. For DR and RetinaMNIST, $k = 5$.

| Algorithm | MORPH II (setting A) | MORPH II (setting B) | DR | RetinaMNIST |
|---|---|---|---|---|
| OL (Lim et al., 2020) | 0.65 | 0.69 | 0.81 | 0.72 |
| Random guess | 1.94 | 1.94 | 1.60 | 1.60 |
| Proposed UOL | 1.20 | 1.31 | 1.08 | 1.00 |

optimally. In contrast, for the proposed UOL algorithm, we do not use the Hungarian method and just reverse the cluster order if it better matches the ground-truth — the reversed order is fundamentally the same as the original order. Nevertheless, UOL significantly outperforms all conventional algorithms in all tests, indicating that it can group instances into meaningfully ordered clusters. Especially, UOL achieves high SRCCs of 0.632 and 0.531 at $k = 6$ in settings A and B of MORPH II, respectively, while the second-best MiCE (Tsai et al., 2021) scores 0.260 and 0.271 only.

Figure 4 compares examples of clustering results in setting A of MORPH II at $k = 6$. The conventional algorithms tend to group similar-looking instances, as they focus on reducing intra-class variance. For example, in cluster 3 in Figure 4(a) or in Figure 4(c), background colors or personal appearances affect the clustering. Thus, they fail to cluster instances according to their ages. On the contrary, the proposed UOL groups more diversely looking people in the same cluster, but those people have similar ages. Also, UOL arranges the clusters well according to the average ages, indicating that UOL can discover the underlying order of instances without any supervision.

Table 2 shows the results on CLAP2015, which is more challenging than MORPH II because it consists of natural face shots in diverse environments. Nevertheless, UOL surpasses the conventional algorithms in all tests.

**Comparison on medical assessment datasets:** Table 3 compares the results on the DR and RetinaMNIST datasets. Note that DeepCluster (Caron et al., 2018), IDFD (Tao et al., 2021), and ProPos (Huang et al., 2022) exploit the ordinary $k$-means for clustering. However, for ordered clustering, the ordinary $k$-means is not optimal because it does not consider the order of clusters. In contrast, based on the ordered $k$-means, the proposed UOL yields the best scores in all tests.

**Results on facial expression dataset:** We test the proposed UOL on the FER+ dataset, containing facial images with various emotions. Figure 5 shows clustering examples of the *Anger*, *Fear*, and *Disgust* images at $k = 5$. Note that there is no explicit order between these three emotion classes. Nevertheless, we see that UOL sorts the images from the strongest (cluster 1) to the weakest (cluster 5) levels of 'making a face.' UOL discovers these ordered groups without any supervision. More results on FER+ are provided in Appendix C.3.

Depending on datasets, UOL may discover different, unexpected orders of clusters. This is a natural property in unsupervised clustering. As long as a discovered order of clusters is meaningful and consistent, it is a good clustering result enabling an initial understanding of data characteristics.

**Embedding spaces:** Figure 6 compares the embedding spaces for DR at $k = 5$, where t-SNE (Maaten & Hinton, 2008) is used for the space visualization. In Figure 6(a)~(d), the conventional algorithms (Caron et al., 2018; Tsai et al., 2021; Tao et al., 2021; Huang et al., 2022) fail to group instances according to their classes. In contrast, UOL sorts the instances reliably, though not perfectly, in Figure 6(e) by employing the SL loss and the deviation loss during the embedding space construction. More comparison results of embedding spaces are provided in Appendix C.6.

Table 5: Ablation studies for the loss function in equation 11 on MORPH II, DR, and RetinaMNIST.

| Method | $\ell_{\mathrm{SL}}$ | $\ell_{\mathrm{D}}$ | MORPH II (A), $k = 6$ | | DR, $k = 5$ | | RetinaMNIST, $k = 5$ | |
|---|---|---|---|---|---|---|---|---|
| | | | SRCC ($\uparrow$) | MAE ($\downarrow$) | SRCC ($\uparrow$) | MAE ($\downarrow$) | SRCC ($\uparrow$) | MAE ($\downarrow$) |
| I | ✓ | | 0.423 | 1.469 | 0.304 | 1.236 | 0.533 | 0.994 |
| II | | ✓ | 0.203 | 1.695 | 0.085 | 1.506 | 0.527 | 0.994 |
| III | ✓ | ✓ | 0.632 | 1.122 | 0.333 | 1.219 | 0.567 | 0.953 |

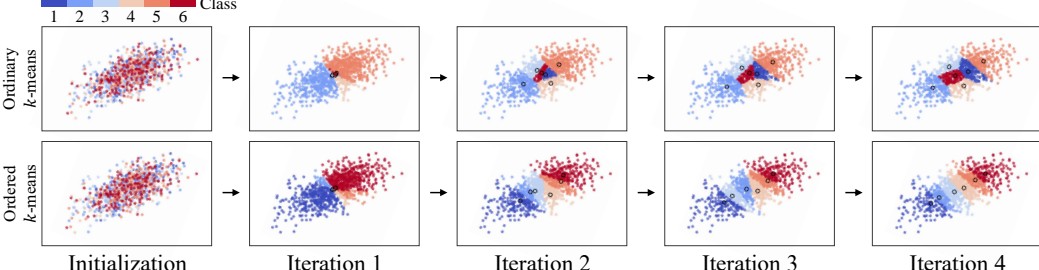

Figure 7: Illustration of the clustering processes of the ordinary $k$-means and the ordered $k$-means.

## 4.4 UNSUPERVISED RANK ESTIMATION

The proposed UOL can estimate the rank of an unseen test instance via equation 12. Table 4 compares the rank estimation results, measured by MAEs, on MORPH II, DR, and RetinaMNIST. To the best of our knowledge, there is no unsupervised rank estimator with reliable source codes. Hence, we provide the results of OL (Lim et al., 2020), which is one of the state-of-the-art supervised rank estimators, as the performance upper bounds. Also, we list the results of the random guess scheme as the lower bounds. We see that UOL provides decent rank estimation results, even though it uses no annotations for both training and testing.

## 4.5 ANALYSIS

**Ablation study:** Table 5 compares ablated methods for the loss function in equation 11. Method I employs the SL loss $\ell_{\mathrm{SL}}$ only, and method II does the deviation loss $\ell_{\mathrm{D}}$ only. Compared with III (UOL), both methods I and II degrade the performances. Especially, by comparing II and III, we see that $\ell_{\mathrm{SL}}$ significantly improves the results. It helps to sort instances and cluster centroids compactly around a chain of line segments in the embedding space.

**Comparison with ordinary $k$-means:** Figure 7 illustrates the clustering processes of the proposed ordered $k$-means and the ordinary $k$-means for a 2D example. As the iteration goes on, the ordinary $k$-means updates the clusters so that the variance of instances in each cluster is reduced. Consequently, instances are located compactly around the centroids, but the clusters are not arranged in an order. In contrast, the ordered $k$-means reduces the deviation of instances from their NCCs. Hence, consecutive clusters are located closely to each other, so they form an order in the embedding space. More result is provided in Figure 11 in Appendix C.6.

## 5 CONCLUSIONS

The UOL algorithm for unsupervised clustering and rank estimation of orderable data was proposed in this work. First, we group instances into ordered clusters via the ordered $k$-means. Then, using the SL loss and the deviation loss, we fine-tune the encoder to revamp the embedding space, so that instances are sorted according to their cluster indexes. The clustering and the feature embedding are jointly optimized. Furthermore, we estimate the rank of an unseen test instance by performing the $K$-NN search in the embedding space without any supervision. Extensive experiments on various orderable datasets demonstrated that UOL provides promising clustering and rank estimation results.

ACKNOWLEDGMENTS

This work was supported by the National Research Foundation of Korea (NRF) grants funded by the Korea government (MSIT) (No. NRF-2021R1A4A1031864 and NRF-2022R1A2B5B03002310).

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

# A  Broader Impacts

Recently, ethical concerns about deep-learning-based systems have been raised (Castelvecchi, 2020; Roussi, 2020; Noorden, 2020). Although the proposed algorithm groups instances, including faces, in an unsupervised manner, the results should never be misinterpreted in such a way as to encourage any kind of discrimination. We recommend using the proposed algorithm for research only.

# B  Algorithm Details

## B.1  Optimality of Centroid Rule

After fixing clusters, we update the centroids to minimize the objective function $J$ in equation 3. For $j \notin \{1, 2, k-1, k\}$, by differentiating $J$ with respect to $\mu_j$ and setting it to zero, we have

$$\frac{\partial J}{\partial \mu_j} = \sum_{x \in \mathcal{C}_j} 2(\mu_j - h_x) + \alpha \sum_{x \in \mathcal{C}_{j_\mathrm{p}}} 2(\mu_j - h_x) + \alpha \sum_{x \in \mathcal{C}_{j_\mathrm{n}}} 2(\mu_j - h_x) \tag{15}$$

$$= 2\big(|\mathcal{C}_j| + \alpha|\mathcal{C}_{j_\mathrm{p}}| + \alpha|\mathcal{C}_{j_\mathrm{n}}|\big)\mu_j - 2\big(\sum_{x \in \mathcal{C}_j} h_x + \alpha \sum_{x \in \mathcal{C}_{j_\mathrm{p}}} h_x + \alpha \sum_{x \in \mathcal{C}_{j_\mathrm{n}}} h_x\big) \tag{16}$$

$$= 0. \tag{17}$$

Therefore, the optimal centroid is given by

$$\mu_j = \frac{\sum_{x \in \mathcal{C}_j} h_x + \alpha \sum_{x \in \mathcal{C}_{j_\mathrm{p}}} h_x + \alpha \sum_{x \in \mathcal{C}_{j_\mathrm{n}}} h_x}{|\mathcal{C}_j| + \alpha|\mathcal{C}_{j_\mathrm{p}}| + \alpha|\mathcal{C}_{j_\mathrm{n}}|} \qquad \text{for } j \notin \{1, 2, k-1, k\}. \tag{18}$$

For $j \in \{1, 2, k-1, k\}$, the centroid rule is slightly different due to endpoint effects. For $j = 1$,

$$\frac{\partial J}{\partial \mu_j} = \sum_{x \in \mathcal{C}_j} 2(\mu_j - h_x) + \alpha \sum_{x \in \mathcal{C}_{j_\mathrm{n}}} 2(\mu_j - h_x) \tag{19}$$

$$= 2\big(|\mathcal{C}_j| + \alpha|\mathcal{C}_{j_\mathrm{n}}|\big)\mu_j - 2\big(\sum_{x \in \mathcal{C}_j} h_x + \alpha \sum_{x \in \mathcal{C}_{j_\mathrm{n}}} h_x\big) \tag{20}$$

$$= 0. \tag{21}$$

Different from equation 15, there is no term related to $j_\mathrm{p}$ in equation 19, since there is no previous cluster for the first cluster $\mathcal{C}_1$. Thus, the optimal centroid is given by

$$\mu_j = \frac{\sum_{x \in \mathcal{C}_j} h_x + \alpha \sum_{x \in \mathcal{C}_{j_\mathrm{n}}} h_x}{|\mathcal{C}_j| + \alpha|\mathcal{C}_{j_\mathrm{n}}|} \qquad \text{for } j = 1. \tag{22}$$

Similarly, we have

$$\mu_j = \frac{\sum_{x \in \mathcal{C}_j} h_x + \alpha \sum_{x \in \mathcal{C}_{j_\mathrm{p}}} h_x}{|\mathcal{C}_j| + \alpha|\mathcal{C}_{j_\mathrm{p}}|} \qquad \text{for } j = k. \tag{23}$$

For $j = 2$, we have

$$\frac{\partial J}{\partial \mu_j} = \sum_{x \in \mathcal{C}_j} 2(\mu_j - h_x) + 2\alpha \sum_{x \in \mathcal{C}_{j_\mathrm{p}}} 2(\mu_j - h_x) + \alpha \sum_{x \in \mathcal{C}_{j_\mathrm{n}}} 2(\mu_j - h_x) \tag{24}$$

$$= 2\big(|\mathcal{C}_j| + 2\alpha|\mathcal{C}_{j_\mathrm{p}}| + \alpha|\mathcal{C}_{j_\mathrm{n}}|\big)\mu_j - 2\big(\sum_{x \in \mathcal{C}_j} h_x + 2\alpha \sum_{x \in \mathcal{C}_{j_\mathrm{p}}} h_x + \alpha \sum_{x \in \mathcal{C}_{j_\mathrm{n}}} h_x\big) \tag{25}$$

$$= 0. \tag{26}$$

Hence, the optimal centroid is

$$\mu_j = \frac{\sum_{x \in \mathcal{C}_j} h_x + 2\alpha \sum_{x \in \mathcal{C}_{j_\mathrm{p}}} h_x + \alpha \sum_{x \in \mathcal{C}_{j_\mathrm{n}}} h_x}{|\mathcal{C}_j| + 2\alpha |\mathcal{C}_{j_\mathrm{p}}| + \alpha |\mathcal{C}_{j_\mathrm{n}}|} \qquad \text{for } j = 2. \tag{27}$$

Similarly,

$$\mu_j = \frac{\sum_{x \in \mathcal{C}_j} h_x + \alpha \sum_{x \in \mathcal{C}_{j_\mathrm{p}}} h_x + 2\alpha \sum_{x \in \mathcal{C}_{j_\mathrm{n}}} h_x}{|\mathcal{C}_j| + \alpha |\mathcal{C}_{j_\mathrm{p}}| + 2\alpha |\mathcal{C}_{j_\mathrm{n}}|} \qquad \text{for } j = k - 1. \tag{28}$$

### B.2 OPTIMALITY OF NCC RULE

Suppose that instance $x$ is declared to belong to cluster $\mathcal{C}_j$. It then contributes to the objective function $J$ in equation 3 by $\delta(h_x, \mu_{j_\mathrm{p}}, \mu_j, \mu_{j_\mathrm{n}})$. If it is assigned to another cluster $\mathcal{C}_l$, then its contribution becomes $\delta(h_x, \mu_{l_\mathrm{p}}, \mu_l, \mu_{l_\mathrm{n}})$. Therefore, to minimize $J$, $x$ should be assigned to $\mathcal{C}_j$ only if

$$\delta(h_x, \mu_{j_\mathrm{p}}, \mu_j, \mu_{j_\mathrm{n}}) \leq \delta(h_x, \mu_{l_\mathrm{p}}, \mu_l, \mu_{l_\mathrm{n}}) \qquad \text{for all } 1 \leq l \leq k. \tag{29}$$

Equivalently, we have the NCC rule in equation 5.

### B.3 PERMUTATION RULE

After fixing the clusters and the centroids, we rearrange the clusters by applying the optimal permutation $\sigma^*$ for minimizing $J$ in equation 3. Note that $\frac{k!}{2}$ permutations should be considered to find the optimal $\sigma^*$, which may require too many computations for a large $k$. Hence, for $k > 6$, we use the following greedy search scheme for $\sigma$.

Let $\sigma_t : \{1, \ldots, k\} \to \{0, 1, \ldots, t\}$ be a partial permutation function. It maps $t$ different indices to $\{1, \ldots, t\}$ bijectively and maps the other $k - t$ indices to zero. Then, the optimal partial permutation $\sigma_t^*$ is determined by

$$\sigma_t^* = \arg\min_{\sigma_t} J(\{\mathcal{C}_{\sigma_t(i)}\}_{i=1}^k, \{\mu_{\sigma_t(i)}\}_{i=1}^k). \tag{30}$$

After obtaining $\sigma_t^*$, we find the optimal $\sigma_{t+1}^* : \{1, \ldots, k\} \to \{0, 1, \ldots, t+1\}$. To this end, we consider two extended permutations $\sigma_{t+1}^\mathrm{n}$ and $\sigma_{t+1}^\mathrm{P}$, satisfying the constraints:

$$\sigma_{t+1}^\mathrm{n}(j) = \sigma_t^*(j) \qquad \text{if } \sigma_t^*(j) \neq 0, \tag{31}$$

$$\sigma_{t+1}^\mathrm{P}(j) = \sigma_t^*(j) + 1 \qquad \text{if } \sigma_t^*(j) \neq 0. \tag{32}$$

Note that $\sigma_{t+1}^\mathrm{n}$ and $\sigma_{t+1}^\mathrm{P}$ preserve the order of the $t$ bijectively mapped indices in $\sigma_t^*$. We select $\sigma_{t+1}$, which minimizes $J$ among all possible $\sigma_{t+1}^\mathrm{n}$ and $\sigma_{t+1}^\mathrm{P}$, as the optimal $\sigma_{t+1}^*$. Starting with $t = 3$, we repeat this process until $\sigma_k^*$ is obtained. Thus, the number of permutations to consider is $\frac{k(k-1)(k-2)}{2} + (k - 3)(k - 2)$, and the complexity is $O(k^3)$.

## C  MORE EXPERIMENTS

### C.1  IMPLEMENTATION DETAILS

We do all experiments using PyTorch (Paszke et al., 2019) and an NVIDIA GeForce RTX 3090 GPU. For the MORPH II dataset, we use settings A and B, which are widely used for evaluation.

- Setting A – 5,492 images of the Caucasian race are selected and then randomly divided into two disjoint subsets: 80% for training and 20% for testing.
- Setting B – 21,000 images of Africans and Caucasians are selected to satisfy two constraints: the ratio between Africans and Caucasians should be $1:1$, and that between females and males should be $1:3$. They are split into three disjoint subsets S1, S2, and S3. We use S2 for training and S1 + S3 for testing.

For clustering on MOPRH II and CLAP, we partition the entire age range into $k$ consecutive age groups. Table 6 lists the age groups at different $k$'s. These age groups are determined so as to contain similar numbers of instances.

Table 6: Partitioning of the entire age range into age groups. The maximum age of each age group is reported. For example, at $k = 6$, the second age group in setting A of MORPH II is (21, 27].

|  | $k = 6$ | $k = 9$ | $k = 12$ |
|---|---|---|---|
| MORPH II (A) | 21 / 27 / 34 / 38 / 43 / 77 | 19 / 22 / 26 / 31 / 34 / 36 / 39 / 42 / 77 | 18 / 20 / 23 / 26 / 30 / 33 / 35 / 37 / 39 / 41 / 43 / 77 |
| MORPH II (B) | 20 / 25 / 32 / 36 / 41 / 77 | 18 / 21 / 25 / 29 / 33 / 36 / 39 / 42 / 77 | 18 / 20 / 22 / 24 / 27 / 30 / 33 / 35 / 37 / 39 / 41 / 77 |
| CLAP | 20 / 23 / 26 / 30 / 37 / 82 | 18 / 21 / 23 / 25 / 27 / 29 / 33 / 38 / 82 | - |

## C.2 MORE RESULTS ON MORPH II

Figure 8 shows examples of clustering results in setting B of MORPH II at $k = 6$. We see that the proposed UOL algorithm groups the instances according to their ages reliably.

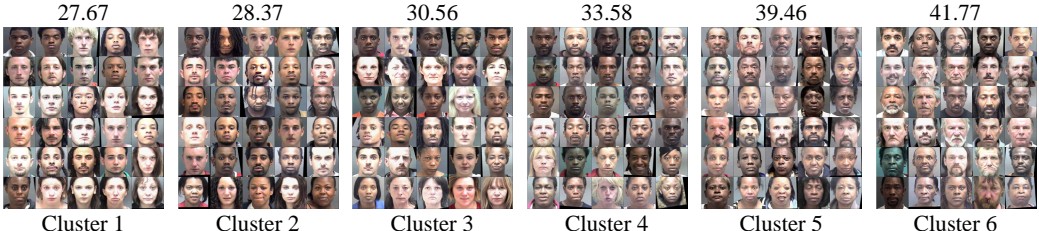

Figure 8: Examples of clustering results on MORPH II (setting B). For each cluster, the average ground-truth age of its instances is also reported.

## C.3 MORE RESULTS ON FER+

Figure 9 shows clustering examples on the FER+ dataset. As done in Figure 5, Figure 9(a) illustrates how the *Anger*, *Fear*, and *Disgust* images are sorted according to the levels of 'making a face.' Figure 9(b) shows clustering results of the *Happy, Neutral,* and *Sad* images, in which the images are roughly sorted as

$$Happy \rightarrow Neutral \rightarrow Sad$$

from cluster 1 to cluster 5. It is worth pointing out that the expression labels are not used in the clustering. Figure 9(c) shows clustering results on the *Surprise* and *Neutral* images. They are sorted from *Neutral* (cluster 1) to *Surprise* (cluster 5).

## C.4 RESULTS ON LANGUAGE AND SPEECH DATASETS

We evaluate the performances of UOL on simple language and speech datasets.

- **French sentiment** (Nguyen, 2022): It is a dataset for regressing sentiment levels in sentences. It offers 944 sentences in French. Each sentence is annotated with a 5-scale sentiment score (1 - very negative, 2 - negative, 3 - neutral, 4 - positive, and 5 - very positive). Figure 10 shows some examples. We adopt BERT (Devlin et al., 2019) as the encoder and use no labels for the UOL training.
- **Speechocean762** (Zhang et al., 2021): It is a dataset for scoring English pronunciation. It contains 5,000 utterances from 250 non-native speakers. Each utterance is annotated with a 10-scale fluency score. We use Wav2Vec2 (Baevski et al., 2020) as the encoder. Only the utterances are used for training.

Table 7 shows that UOL shows promising ordered clustering results on these language and speech datasets, as well as the image datasets.

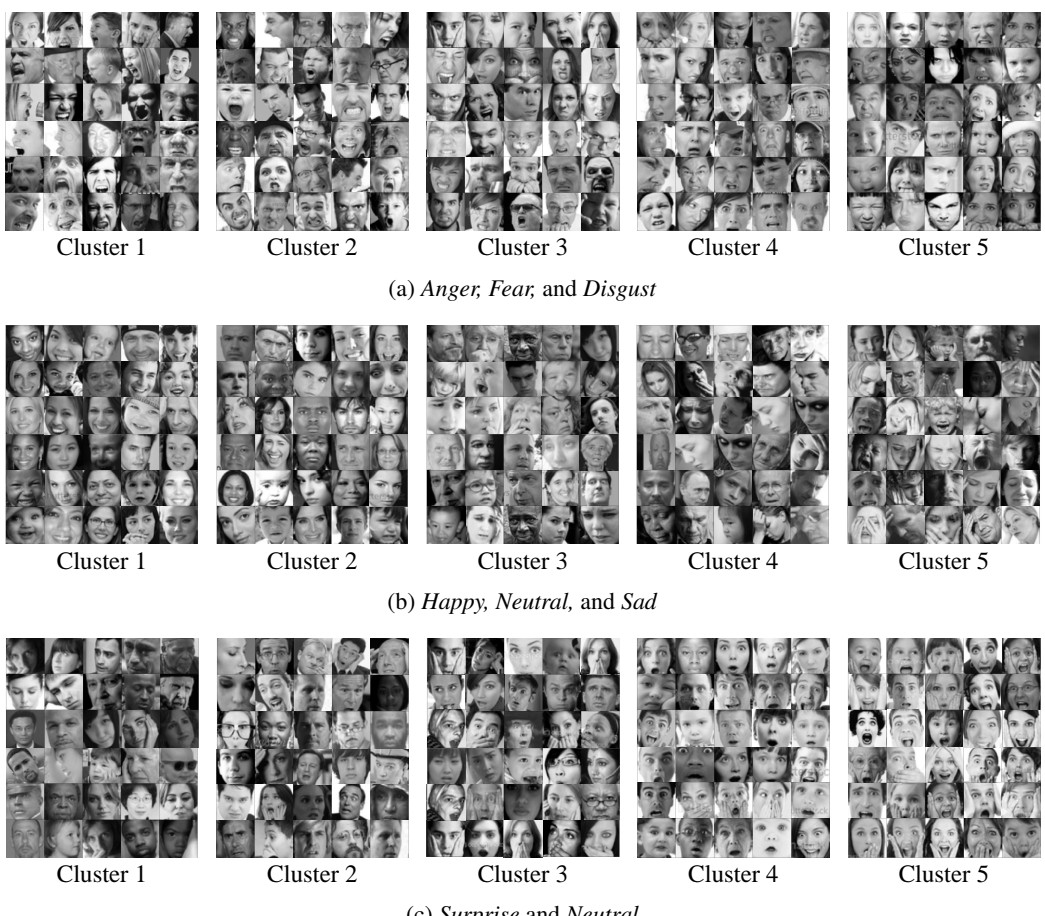

(a) *Anger, Fear,* and *Disgust*

(b) *Happy, Neutral,* and *Sad*

(c) *Surprise* and *Neutral*

Figure 9: Examples of clustering results on the FER+ dataset at $k = 5$.

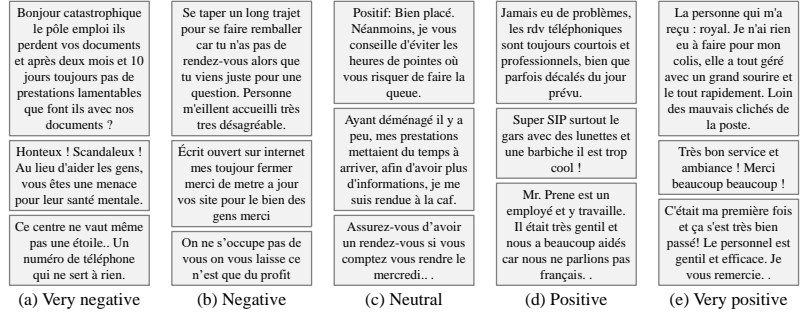

(a) Very negative (b) Negative (c) Neutral (d) Positive (e) Very positive

Figure 10: Examples in the French sentiment dataset.

Table 7: Ordered clustering results on the French sentiment and Speechocean762 datasets at $k = 5$.

| Algorithm | French sentiment | | Speechocean762 | |
|---|---|---|---|---|
| | SRCC($\uparrow$) | MAE ($\downarrow$) | SRCC($\uparrow$) | MAE ($\downarrow$) |
| MiCE (Tsai et al., 2021) | 0.196 | 1.129 | 0.139 | 1.314 |
| IDFD (Tao et al., 2021) | 0.063 | 1.258 | 0.105 | 1.380 |
| ProPos (Huang et al., 2022) | 0.042 | 1.326 | 0.107 | 1.379 |
| Proposed UOL | **0.448** | **0.989** | **0.672** | **0.991** |

## C.5 UNSUPERVISED RANK ESTIMATION

Table 8 is an extended version of the unsupervised rank estimation results in Table 4. We measure the performances of the conventional algorithms (Caron et al., 2018; Tao et al., 2021; Huang et al., 2022) using the simple $K$-NN classification in equation 12, as in UOL. However, since these algorithms do not arrange the clusters meaningfully, their cluster indices contain no rank information. Thus, we permute their clusters optimally via the Hungarian method (Kuhn, 1955). On the contrary, the proposed UOL requires no such supervision. Nevertheless, UOL outperforms the conventional algorithms meaningfully in all tests.

Table 8: Comparison of rank estimation results (MAEs) on the MORPH II, DR, and RetinaMNIST datasets. For MORPH II, $k = 6$. For DR and RetinaMNIST, $k = 5$.

| Algorithm | MORPH II (A) | MORPH II (B) | DR | RetinaMNIST |
|---|---|---|---|---|
| Random guess | 1.94 | 1.94 | 1.60 | 1.60 |
| DeepCluster (Caron et al., 2018) | 1.59 | 1.73 | 1.29 | 1.16 |
| IDFD (Tao et al., 2021) | 1.57 | 1.84 | 1.32 | 1.43 |
| ProPos (Huang et al., 2022) | 1.43 | 1.54 | 1.25 | 1.28 |
| Proposed UOL | **1.20** | **1.31** | **1.08** | **1.00** |

## C.6 MORE ANALYSIS

**Analysis on $\alpha$:** Table 9 compares the clustering results with different $\alpha$'s. Note that $\alpha$ is a positive weight to control the influence of adjacent centroids in equation 1. It is reasonable to set $\alpha$ to be less than 0.5. Otherwise, the distances to adjacent centroids affect the objective function more greatly than the distance to the corresponding centroid does.

In all tests, we fix $\alpha = 0.2$, which provides the best results in Table 9. However, at both $\alpha = 0.1$ and $\alpha = 0.3$, UOL still shows comparable or better results than the conventional algorithms in Tables 1 and 3. At $\alpha = 0$, the ordered $k$-means becomes the ordinary $k$-means. In such a case, the performances degrade severely, for the clustering does not consider the order of clusters.

Table 9: Comparison of ordered clustering results on MORPH II (setting A) and RetinaMNIST according to $\alpha$ in equation 1. For MORPH II, $k = 6$. For RetinaMNIST, $k = 5$.

| Dataset | $\alpha = 0$ | | $\alpha = 0.1$ | | $\alpha = 0.2$ | | $\alpha = 0.3$ | |
| | SRCC ($\uparrow$) | MAE ($\downarrow$) | SRCC ($\uparrow$) | MAE ($\downarrow$) | SRCC ($\uparrow$) | MAE ($\downarrow$) | SRCC ($\uparrow$) | MAE ($\downarrow$) |
|---|---|---|---|---|---|---|---|---|
| MORPH II (A) | 0.159 | 1.673 | 0.540 | 1.267 | 0.632 | 1.122 | 0.479 | 1.339 |
| RetinaMNIST | 0.106 | 1.311 | 0.544 | 0.989 | 0.567 | 0.953 | 0.522 | 1.023 |

**Analysis on $\gamma$:** Table 10 compares the performances at different $\gamma$'s. In equation 10, $\gamma$ is a threshold to suppress the deviation $\delta$ of an instance from its NCCs to be small. If $\gamma$ is too small, clusters may highly overlap with each other in the embedding space. On the other hand, if $\gamma$ is too large, instances may not be located compactly around their corresponding centroids. In Table 10, the best results are achieved at $\gamma = 0.25$. Therefore, we fix $\gamma = 0.25$ in all tests.

Table 10: Comparison of ordered clustering results on MORPH II (setting A) and RetinaMNIST according to $\gamma$. We set $k = 6$ and $k = 5$ for MORPH II and RetinaMNIST, respectively.

| Dataset | $\gamma = 0.2$ | | $\gamma = 0.25$ | | $\gamma = 0.3$ | | $\gamma = 0.35$ | |
| | SRCC ($\uparrow$) | MAE ($\downarrow$) | SRCC ($\uparrow$) | MAE ($\downarrow$) | SRCC ($\uparrow$) | MAE ($\downarrow$) | SRCC ($\uparrow$) | MAE ($\downarrow$) |
|---|---|---|---|---|---|---|---|---|
| MORPH II (A) | 0.543 | 1.275 | 0.632 | 1.122 | 0.592 | 1.180 | 0.567 | 1.220 |
| RetinaMNIST | 0.528 | 1.002 | 0.567 | 0.953 | 0.532 | 0.991 | 0.541 | 0.970 |

**More comparison with ordinary $k$-means:** Figure 11 compares the clustering results of the ordinary $k$-means and the ordered $k$-means on 1D manifold data. The data points are sampled around an S curve in the 3D space. The clusters of the ordinary $k$-means are not arranged in an explicit order. In contrast, the ordered $k$-means groups the data points into sequentially ordered clusters along the 1D manifold structure.

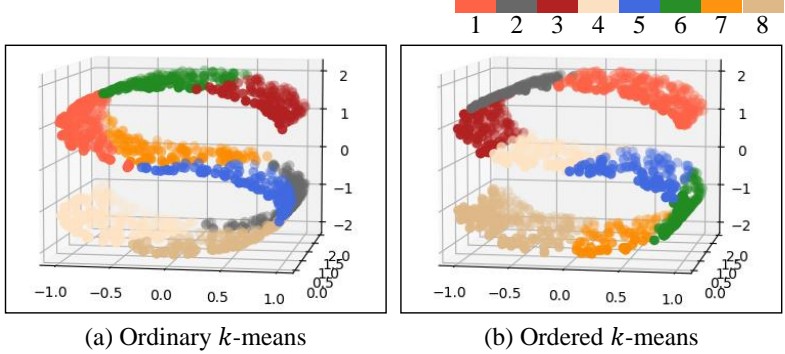

(a) Ordinary $k$-means        (b) Ordered $k$-means

Figure 11: Comparison of clustering results.

**Comparison of embedding spaces:** Figure 12 is an extended version of Figure 1(b), which compares the embedding spaces for MORPH II (setting A). The conventional algorithms cluster object instances in different classes closely without clear distinction; they fail to discover underlying classes reliably. In contrast, UOL well arranges the instances according to their classes in the embedding space.

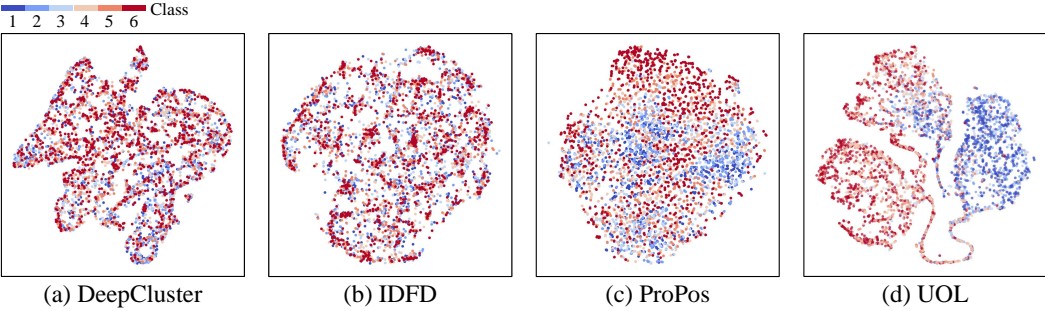

(a) DeepCluster      (b) IDFD      (c) ProPos      (d) UOL

Figure 12: Comparison of the embedding spaces, for dividing MORPH II (setting A) into six clusters, obtained by (a) DeepCluster (Caron et al., 2018), (b) IDFD (Tao et al., 2021), (c) ProPos (Huang et al., 2022), and (d) the proposed UOL algorithm. We group the ground-truth age classes into six age ranges, depicted by colors from blue to red.

**Embedding space transition:** Figure 13 visualizes the transition of the embedding space for MORPH II (setting A) at $k = 6$. As the training step increases, instances are gradually sorted to satisfy the order desideratum in equation 7. Hence, after the training, the instances in different classes are more clearly divided in the embedding space.

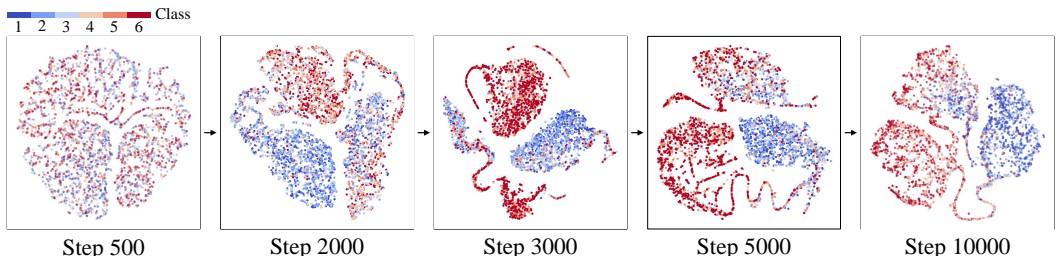

Step 500      Step 2000      Step 3000      Step 5000      Step 10000

Figure 13: Embedding space transition for MORPH II (setting A) during the UOL training at $k = 6$.

**Unordered metrics:** In Table 11, we compare clustering results by employing two unordered metrics: Accuracy and normalized mutual information (NMI). Compared to ordinary classification datasets, it is more difficult to group instances according to their classes in these orderable datasets of MORPH II and RetinaMNIST. This is because adjacent classes are not clearly distinguished. For example, 16-year-olds and 20-year-olds may have similar appearances. Therefore, the overall scores are relatively low. However, the proposed UOL performs the best in all tests even in terms of the unordered metrics.

Table 11: Comparison of clustering results on MORPH II (setting A) and RetinaMNIST in terms of two unordered metrics: Accuracy and NMI. For MORPH II, $k = 6$. For RetinaMNIST, $k = 5$.

| Algorithms | MORPH II (A) | | RetinaMNIST | |
|---|---|---|---|---|
| | Accuracy (%) ($\uparrow$) | NMI ($\uparrow$) | Accuracy (%) ($\uparrow$) | NMI ($\uparrow$) |
| MiCE (Tsai et al., 2021) | 24.1 | 0.046 | 32.3 | 0.104 |
| IDFD (Tao et al., 2021) | 21.9 | 0.007 | 29.3 | 0.098 |
| ProPos (Huang et al., 2022) | 20.4 | 0.014 | 31.2 | 0.105 |
| Proposed UOL | **31.0** | **0.154** | **37.9** | **0.151** |

**Impacts of different backbones:** Table 12 lists the ordered clustering results using different backbones on the MORPH II (setting A) and RetinaMNIST datasets. UOL also performs well with these backbones, indicating that it is not affected by the backbones too much.

Table 12: Comparison of ordered clustering results with different backbones on the MORPH II (setting A) and RetinaMNIST datasets. For MORPH II, $k = 6$. For RetinaMNIST, $k = 5$.

| Backbone | MORPH II (A) | | RetinaMNIST | |
|---|---|---|---|---|
| | SRCC($\uparrow$) | MAE ($\downarrow$) | SRCC($\uparrow$) | MAE ($\downarrow$) |
| VGG16 | 0.632 | 1.122 | 0.567 | 0.953 |
| ResNet18 | 0.540 | 1.246 | 0.515 | 0.989 |
| ResNet50 | 0.590 | 1.179 | 0.559 | 0.967 |

**Impacts of cluster initialization:** In Table 13, we list the SRCC results of five random cluster initializations. Also, we report the average and standard deviation of the SRCC scores. UOL yields roughly the same order regardless of the cluster initialization, indicating that it is not very sensitive to the initialization.

Table 13: Comparison of the SRCC scores with five random initializations on the MORPH II (setting A) and RetinaMNIST datasets. For MORPH II, $k = 6$. For RetinaMNIST, $k = 5$.

| | 1 | 2 | 3 | 4 | 5 | Average |
|---|---|---|---|---|---|---|
| MORPH II(A) | 0.632 | 0.625 | 0.630 | 0.634 | 0.619 | 0.628±0.005 |
| RetinaMNIST | 0.567 | 0.564 | 0.552 | 0.567 | 0.567 | 0.563±0.006 |

**Limitations:** In Table 2, UOL fails to yield decent clustering results on the CLAP dataset, even though it shows good results on the MORPH II dataset. It is because CLAP contains more diverse face shots in various environments than MORPH II, as illustrated in Figure 14. Also, in addition to facial age estimation and medical assessment, aesthetic score regression is a popular ranking task. However, even for UOL, it is hard to learn the aesthetic criteria in an unsupervised manner, which are often subjective and ambiguous.

Since UOL is the first attempt at ordered clustering, we focus on developing a simple but reasonable algorithm. Hence, instead of considering all possible cases, UOL assumes that there is a total order of underlying classes. However, this is not always the case. Therefore, in order to broaden the potential use cases, it is required to extend the proposed algorithm to the scenarios where there is no total order *e.g.* dataset with a partial order. We leave this for future work.

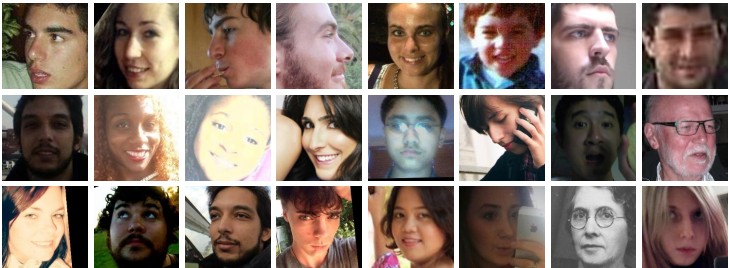

Figure 14: Examples of facial photos in the CLAP dataset.

