# OpenReview forum: "Unsupervised Order Learning"
_ICLR.cc/2024/Conference — ICLR 2024 poster_

### Official Review · Reviewer_rbKP · 2023-10-29

**Soundness:** 3 good
**Presentation:** 3 good
**Contribution:** 2 fair
**Rating:** 5
**Confidence:** 5

**Summary:**

This paper proposed an unsupervised order clustering algorithm for dealing with order data To be specific, authors first proposed a ordered k-means algorithm which defines a measurement of the deviation of sample x from the cluster centroid chain in learned embedding space. Then authors claim that the ordered clustering can be defined as the distance between sample x and its centroid and two neighbored centroids. Based on these, authors proposed an ordered k-means algorithm for clustering ordered data.

**Strengths:**

This paper is well-written and the core idea and motivation are easy to follow.

**Weaknesses:**

First, In page 3, authors claimed that “we propose the first unsupervised algorithm for order learning”. Actually, this paper belongs to a kind of ordered data clustering task, which has been studied in many previous works, such as
[1] An ordinal data clustering algorithm with automated distance learning, AAAI, 2020;
[2] Deep repulsive clustering of ordered data based on order-identity decomposition, ICML, 2020.
Thus, this sentence is not precise.

Second, the deviation of sample x from the chain is borrowed from Lim et al. 2020, thus the true contribution of this paper is the ordered k-means algorithm in Algorithm 1 which is easy to deduce if we have Eq. (1). In a word, I think the true contribution is not enough for ICLR.

Thiredly, in experiments, the compared algorithms mostly are traditional clustering algorithms which can not verify the effectiveness of proposed methods.

In all, I prefer to give the “marginally below the acceptance threshold” decision.

**Questions:**

N/A

---

> ### Author Response · Authors · 2023-11-11
> **Rebuttal by Authors**
>
> Thank you for your constructive comments. We have revised the paper to address your comments faithfully. We have highlighted the revised parts in blue. Please find our responses below.
>
> ---
>
> > **Relation with DLC (Zhang et al., AAAI 2020)**
>
> DLC is an *ordinary* clustering algorithm, but it attempts to cluster data with ordinal attribute values. Whereas ordinary clustering algorithms, including DLC, do not consider the order between resultant clusters, the proposed *ordered* clustering attempts to find clusters together with their meaningful order.
>
> For example, DLC aims to group car instances with (already provided) ordinal attributes, such as buying cost, safety score, maintenance cost, and the number of doors.  A buying cost is labeled as one of four ordinal categories: ‘low,’ ‘median,’ ‘high,’ and ‘very high.’ The difference between ‘low’ and ‘median’ may not be the same as that between ‘high’ and ‘very high.’ DLC attempts to properly define the distance between these attribute values for better clustering. It is worth pointing out that DLC requires data instances with ordinal attribute values, which is a very different scenario from the assumption of the proposed UOL.
>
> We have discussed the relation of the proposed UOL with DLC in the revision. Please see the last paragraph on page 2.
>
> > **Relation with DRC-ORID (Lee and Kim, ICLR 2021)**
>
> DRC-ORID is an unsupervised clustering algorithm to group instances according to the properties unrelated to order (e.g. ethnicity and gender). To this end, Lee and Kim assumed that all ordering relationships between instances --- which are what the proposed UOL aims to discover --- are known. For example, DRC-ORID aims to group facial images into three categories of ‘African American,’ ‘Caucasian,’ and ‘Asian,’ which are unrelated to ages. In contrast, UOL aims to group instances according to their ages. Therefore, the objective of DRC-ORID is orthogonal to ours. This has been clarified. Please see 3rd paragraph on page 3.
>
> > **Deviation from the chain**
>
> Please note that the deviation of an instance $x$ from consecutive centroids in Eq. (1) is first proposed in this paper. It is not borrowed from OL (Lim et al., ICLR 2020). The unsupervised clustering for the $k$-chain hypothesis in OL assigns an instance $x$ to clusters (chains) based on affinity scores. However, the deviation in Eq. (1) and the affinity scores in OL are not related at all.
>
> Moreover, as in DRC-ORID, the objective of clustering in OL is to group instances into clusters according to order-unrelated properties (e.g. gender in a facial age dataset). In contrast, UOL aims to cluster instances according to order-related properties (e.g. age in a facial age dataset). We have clarified this difference. Please see 3rd paragraph on page 3.
>
> > **Comparison with traditional algorithms**
>
> As mentioned in the above responses, there is no conventional algorithm for ordered clustering. Therefore, we compare our results with the clustering algorithms for nominal data. However, to make the comparison as fair as possible, we permute the clusters of those algorithms via the Hungarian method so that the order of the clusters matches the ground truth as well as possible.
>
> ---
>
> If you have any additional concerns, please let us know. We will address them faithfully. We do appreciate your constructive comments again.

---

### Official Review · Reviewer_T1Ca · 2023-10-31

**Soundness:** 3 good
**Presentation:** 3 good
**Contribution:** 3 good
**Rating:** 6
**Confidence:** 5

**Summary:**

This paper proposes a new algorithm, called unsupervised order learning (UOL), for clustering ordered data. It aims to discover hidden ranks (or ordered classes) of objects with no supervision.

**Strengths:**

1. The author proposes the first deep clustering algorithm for ordered data.

2. The authors have introduced the ordered k-means algorithm, which extends the conventional k-means approach.

3. This enhanced method effectively groups object instances and arranges the resulting clusters in a meaningful order. The authors have also provided a proof of the local optimality of the solution.

**Weaknesses:**

See questions

**Questions:**

1. As a clustering method, it is inappropriate and unfair to compare only two types of metrics regarding the order of the data, SRCC and MAE. Some basic clustering metrics, such as ACC and NMI, lack comparison. Also this explains why other comparison algorithms achieve poorer performance.

2. We question the value of unsupervised order clustering. The important value of clustering as a classical unsupervised method is that it does not require a tedious data preprocessing process such as labeling data in advance. In contrast, the order clustering proposed by the authors has high requirements for the dataset itself (sequentially), and such requirements are usually obtained by tedious manual sorting, which contradicts the advantages of clustering itself. Can the authors provide a real dataset or scenario where sequential order exists and clustering is required? Note that this is different from the manually ordered dataset used by the authors in the experimental section.

3. Why did you select only two data sets for your different ablation experiments? Did the authors artificially select the datasets to present the ablation experiments? Meanwhile, the parameter \gamma lacks ablation experiments with relevant parameter descriptions. More experimental results are expected

---

> ### Author Response · Authors · 2023-11-11
> **Rebuttal by Authors**
>
> We do appreciate your constructive comments. We have addressed them faithfully in the revised paper, and we have marked the revised parts in blue. Please find our responses below.
>
> ---
>
> > **Accuracy & NMI**
>
> We compared the accuracy and NMI scores on the MORPH II and RetinaMNIST datasets. Please see Table 11 and its description on page 19.
>
> As compared to ordinary classification datasets, it is more difficult to group instances according to their classes in these ranking (ordinal classification) datasets. This is because adjacent classes are not clearly distinguished. For example, 16-year-olds and 20-year-olds may have similar appearances. Therefore, the overall scores are relatively low. However, note that the proposed UOL performs the best in all tests.
>
> > **The value of ordered clustering**
>
> Please note that the proposed UOL does not require any additional data preprocessing, labeling, or manual sorting of data. In other words, it uses training instances only, as conventional deep clustering algorithms do. Also, UOL uses pseudo labels for its training, but these labels are estimated via the ordered $k$-means automatically. Using pseudo labels for network training is a common practice in deep clustering.
>
> The primary goal of UOL is to group instances according to their classes, as in ordinary clustering. However, as shown in the experiments on datasets with underlying orders, UOL provides better clustering results than conventional algorithms. Hence, as described in the 4th paragraph in Section 1 on page 1, UOL can reduce the annotation burden, for example, for medical data by generating initial prediction results. Similarly, it can be used for other types of data for ranking tasks, such as facial age estimation and historical image classification. Furthermore, it is shown in Appendix C.4 on page 15 that UOL also yields promising results on a language dataset and an audio dataset, as well as image datasets.
>
> > **Parameter $\gamma$**
>
> We compared the performances according to $\gamma$ in Table 10. As you suggested, we have discussed how to select $\gamma$ in more detail. Please see Table 10 and its description on page 17.
>
> > **Ablation study**
>
> Thank you for your suggestion. Below are additional ablation results on the RetinaMNIST dataset. Similar to MORPH II and DR, RetinaMNIST provides similar ablation results: both ablated methods I and II degrade the performances in comparison with UOL (III). We have added these results in the revised paper. Please see Table 5 on page 9.
>
> | Method | $\ell_{\text{SL}}$ | $\ell_{\text{D}}$ | SRCC   | MAE   |
> |--------|--------------------|-------------------|--------|-------|
> | I      | v                  |                   | 0.533  | 0.994 |
> | II     |                    | v                 | 0.527 | 0.994 |
> | III    | v                  | v                 | 0.567  | 0.953 |
>
> ---
>
> If you have any additional concerns, please let us know. We will address them faithfully. Thank you again for your constructive comments.

---

> > ### Comment · Reviewer_T1Ca · 2023-11-23
> > **Thanks for your response**
> >
> > Thanks for your response! I'd like to keep my positive score and increase the confidence to 5.

---

> > > ### Author Response · Authors · 2023-11-23
> > > **Comments by Authors**
> > >
> > > Thank you for your decision to give a positive score! However, the current score 5 is on the negative side under the ICLR rating system. Could you please check it and update the score accordingly?
> > >
> > > Thank you again for your positive evaluation of our paper.

---

### Official Review · Reviewer_kD3Z · 2023-11-02

**Soundness:** 3 good
**Presentation:** 3 good
**Contribution:** 2 fair
**Rating:** 5
**Confidence:** 3

**Summary:**

The paper presents an unsupervised algorithm for clustering ordered data. Specifically, it proposes a so-called ordered k-means algorithm, in which the rules to update the centroids and to find the assignments are modified by adding some reference terms with respect to the previous cluster and the next cluster. Experiments on benchmark datasets are conducted, showing some improvements over the listed baseline algorithms.

**Strengths:**

+ It sounds interesting to modify the ordinary $k$-means algorithm to handle ordered data clustering.
+ Experimental results on benchmark datasets show promising improvements.

**Weaknesses:**

- The difference from the ordinary $k$-means algorithm is the way to update the mean and the way to assign clustering index, both of the two stages are computed by taking a tradeoff between the current clusters and the socalled previous cluster and the next cluster. However, it is not always meaningful to define the previous cluster and the next cluster, if the dimension of the embedding space  (which is also the dimension of the centriods of the clusters) is larger than 2.

- In Eq. (4), the formula to update the centroids contains a parameter $\alpha$.  From the results in Table 9, the performance is sensitive the parameter $\alpha$. Without the proper value for parameter $\alpha$, the promising performance cannot be obtained.

**Questions:**

- It seems not always meaningful to define the previous cluster and the next cluster, provided that the dimension of the embedding space (which is also the dimension of the centriods of the clusters) is larger than 2.

- In Eq. (4), the formula to update the centroids contains a parameter $\alpha$. As can be read from Table 9, the performance of the clustering is very sensitive the value of the parameter $\alpha$. The promising performance cannot be obtained without using the proper value of $\alpha$. Is there any principled rule to set it? Moreover, does the proper value of $\alpha$ vary from dataset to dataset?

---

> ### Author Response · Authors · 2023-11-11
> **Rebuttal by Authors**
>
> We do appreciate your constructive comments. We have addressed them faithfully in the revised paper. We have highlighted the revised parts in blue. Please find our responses below.
>
> ---
>
> > **Previous and next clusters**
>
> We agree that it is hard to define an order between points in a high-dimensional space in general. However, we employ the straight line loss in Eq. (9), which encourages instances to be located near a 1D manifold in the embedding space. Therefore, we can arrange the clusters according to their order along the 1D manifold in the high-dimensional space. This has been clarified in the revision. Please see the last paragraph on page 5.
>
> > **Sensitivity to $\alpha$**
>
> Please note that we fix $\alpha=0.2$ for all datasets. Even though the performances are affected by $\alpha$, they are not sensitive when $\alpha$ is around 0.2. Moreover, by comparing Table 9 with Tables 1 and 3, we can see that UOL yields better or comparable results than the conventional algorithms at $\alpha = 0.1$ and $\alpha = 0.3$ as well, despite 50% differences from the default $\alpha=0.2$. We have revised the paper to clarify this point. Please see Appendix C.6 on page 17.
>
> Also, we will compare the performances on various datasets by varying $\alpha$ more finely in 0.01 units, instead of 0.1 units.
>
> ---
>
> If you have any additional concerns, please let us know. We will address them faithfully. Thank you again for your constructive comments.

---

> > ### Comment · Reviewer_kD3Z · 2023-11-22
> >
> > The reviewer has read the clarification in the rebuttal but did not find the reason to change the pre-rating.

---

> > > ### Author Response · Authors · 2023-11-22
> > > **Comments by Authors**
> > >
> > > Thank you for your comment on our response. However, could you elaborate which part of the paper that you find not satisfactory? We will do our best to resolve your remaining concerns.

---

### Official Review · Reviewer_cgay · 2023-11-03

**Soundness:** 3 good
**Presentation:** 3 good
**Contribution:** 2 fair
**Rating:** 6
**Confidence:** 4

**Summary:**

The paper introduces an unsupervised method to perform clustering when there exists a total order between clusters. This total order can for instance define the age of people in images, and people who are about the same age tend to be in the same cluster or neighbor clusters. The proposed method is similar to soft clustering in the sense that it assigns samples to different clusters (here, at most 3 clusters) and updates the centroids accordingly. However, it also considers some total order on the centroids.

**Strengths:**

The paper is well-written and the exposition of the method is clear. The difference with standard clustering algorithms that do not consider orders between clusters, and also with other ranking methods that use supervision is well defined. Assuming that there exists an order between the cluster in the dataset, the motivation for using the proposed method is clear.
As motivated in the paper, the method can be used as a pretraining stage for downstream tasks including ordinal regression, or it can facilitate an initial understanding of data characteristics. However, only the latter part is evaluated in the paper.

**Weaknesses:**

One main limitation of the method is that it assumes that there exists a total order between the categories/clusters, which is not always the case. The idea of the paper is similar to relative attributes [A] although it considers the unsupervised case. Even with relative attributes, it may be difficult to define a total order between categories so an equivalence between pairs of categories is sometimes defined. Partially ordered sets are in general easier to define than total orders.

Moreover, there might exist different ways to define orders between categories. For instance, in ref [A], the orders between face categories might define age, chubbiness, roundness, color, big lips etc... Fortunately, in Fig 1 (c) of the submission, the face images are ordered by age, but another criterion might have been extracted by the method since it is unsupervised, and the reported scores would not have been as good.

[A] Devi Parikh & Kristen Grauman, Relative Attributes, ICCV 2011

**Questions:**

How important is initialization? Assuming that there exist different possible orders between categories, would one initialization reflect one order (for instance age) and another reflect something else (for instance color)? And in this case, how would the method be useful for real-world applications since there is no way to control the extracted clustering order? In particular, if we consider network pre-training, one initialization would improve performance only if the extracted order aligns with the target order.

---

> ### Author Response · Authors · 2023-11-11
> **Rebuttal by Authors**
>
> Thank you for your positive review and insightful suggestions. We have revised the paper to address your comments faithfully. We have highlighted the revised parts in blue. Please find our responses below.
>
> ---
>
> > **Assumption of total order**
>
> We agree with you. Please note that UOL is the first attempt to cluster ordered data. Hence, we focus on developing a simple but reasonable algorithm for this task, instead of considering all possible cases. It is a future research issue to extend the proposed algorithm for scenarios where there is no total order. We have clarified this point in the revised paper. Please see 1st paragraph in Section 3 on page 3 and the last paragraph on page 20.
>
> > **Different orders**
>
> We also agree that different orders based on different criteria can be discovered. However, this is a natural property in unsupervised clustering. As long as the discovered order of clusters is meaningful and consistent, we believe that it is a good clustering result, which can facilitate an understanding of data characteristics. This has been clarified in the revision. Please see the second last paragraph on page 8.
>
> Since UOL is an unsupervised learning algorithm, it is likely (and also desirable)  to learn the most dominant and obvious order in a dataset. Thus, in our experiments on facial age datasets, the clusters are divided mainly according to ages, which are the most prominent characteristics of those datasets. Similarly, in our experiments on the FER+ dataset in Figure 5 on page 7, UOL sorts the images according to the level of ‘making a face,’ because it is the dominant and consistent ordering criterion for describing the three different emotions.
>
> > **Initialization**
>
> Table 13 on page 19 lists the SRCC results of five random initializations. In the experiments, UOL yields roughly the same order regardless of the cluster initialization. This is because UOL tends to learn the most obvious order in a dataset, as mentioned above.
>
> ---
>
> If you have additional comments, please let us know. Thank you again for your positive comments.

---

> > ### Comment · Reviewer_cgay · 2023-11-21
> > **Thank you**
> >
> > Thank you for the addition of different initializations. Why did you include only the average performance and not the standard deviation in Table 13?

---

> > > ### Author Response · Authors · 2023-11-21
> > > **We do appreciate your positive comments.**
> > >
> > > Table 13 lists each SRCC score of five random cluster initializations. Hence, the average and standard deviation of the SRCC scores are computed as below.
> > >
> > > | | |
> > > |--------------|-----------------|
> > > | MORPH II (A) | $0.628\pm0.005$ |
> > > | RetinaMNIST  | $0.563\pm0.006$ |
> > >
> > >
> > > Thank you again for your time and effort for reviewing our paper. We do appreciate it. If you have additional comments, please let us know.

---

> > > > ### Author Response · Authors · 2023-11-22
> > > > **Comments by Authors**
> > > >
> > > > We have added the average and standard deviation of SRCC scores in the revised paper. Please see Table 13 on page 19.
> > > >
> > > > ---
> > > >
> > > > Thank you again for your constructive comments.

---

### Author Response · Authors · 2023-11-10
**Author response to all reviewers**

We would like to thank all reviewers for their time and efforts for providing constructive reviews. We will upload our response to each question or comment as soon as possible.

---

### Author Response · Authors · 2023-11-14
**Author response to all reviewers**

We have uploaded the our response to each question or comment. Please find our responses below. If our responses could not resolve your concerns, please let us know. Also, we have revised the paper to address the comments faithfully. We thank all reviewers for their time and efforts for constructive reviews again.

---

### Meta-Review · Area_Chair_hwyy · 2023-12-09

**Metareview:**

**Summary**
This paper explores the clustering of multivariate data, where a total order between clusters is also learned in a fully unsupervised manner. The approach extends the standard $K$-means algorithm, where an encoder is fine-tuned with a carefully designed loss at each iteration. This refinement encourages that data points are linearly arranged on a 1D manifold in the embedding space. The effectiveness of the proposed algorithm, ordered $K$-means, has been demonstrated on synthetic and real-world datasets.

**Strengths**
- The studied problem of ordered $K$-means is simple yet important. There are a number of literature that assumes a predetermined order between data points and use it for clustering, as discussed by the authors in the paper. However, the problem setting of this paper, where a dataset is ordinary multivariate data and the order between clusters is learned in an unsupervised manner, is both interesting and relevant.
- Mathematical formulation and expressions are clear and easy to follow. The proposed algorithm is simple.
- Empirical evaluation is carefully performed and results are convincing.

**Weaknesses**
- The presentation is a bit misleading. Specifically, the authors often say "clustering for ordered data". However, this expression might lead to misunderstanding, as it suggests a predetermined order on the data space, which is not the focus of this paper and actually results in misunderstanding of the reviewers. I strongly recommend carefully revising the text to clearly distinguish "order for clusters that will be learned" (the scope of the paper) and "order for data points that is known in advance" (not addressed in the paper).
- As Reviewer cgay pointed out, total orders are a bit too restrictive and partial orders are more flexible and fundamental. I would appreciate seeing an  extension of this paper to partial order $K$-means, which would allow the embedding of data points onto multi-dimensional manifolds instead of just 1D. I believe this extension could be a substantial contribution to the community.

**Justification For Why Not Higher Score:**

There are still some weaknesses as I have listed in the meta-review, and some presentation requires minor revision. Considering my opinion and the reviewers' scores, I believe "Accept (poster)" is appropriate for this paper.

**Justification For Why Not Lower Score:**

Reviewers raised concerns mainly about the motivation of (total) orders and their relationship to existing literature. However, all major concerns have been  addressed in the authors' response. Additionally, after carefully reading the paper and reviewers' comments, I believe the contribution of this paper is strong and the paper is well-written. Therefore I would like to recommend accepting the paper.

---

### Decision · Program_Chairs · 2024-01-16

Accept (poster)